# AN $O(k \log n)$ TIME FOURIER SET QUERY ALGORITHM

## ABSTRACT

Fourier transformation is an extensively studied problem in many research fields. It has many applications in machine learning, signal processing, compressed sensing, and so on. In many real-world applications, approximated Fourier transformation is sufficient and we only need to do the Fourier transform on a subset of coordinates. Given a vector $x \in \mathbb{C}^n$, approximation parameters $\epsilon, \delta \in (0, 0.1)$, and a query set $S \subset [n]$ of size $k$, we propose an algorithm to compute an approximate Fourier transform result $x'$ which uses $O(\epsilon^{-1} k \log(n/\delta))$ Fourier measurements and runs in $O(\epsilon^{-1} k \log(n/\delta))$ time. For $\widehat{x}$ being the Fourier transformation result, our algorithm can output a vector $x'$ such that $\|(x' - \widehat{x})_S\|_2^2 \leq \epsilon \|\widehat{x}_{\overline{S}}\|_2^2 + \delta \|\widehat{x}\|_1^2$ holds with probability of at least $9/10$, where $\overline{S}$ denotes the complement of the set $S$, i.e., $\overline{S} := [n] \setminus S$.

## 1    INTRODUCTION

Fourier transform is ubiquitous in image and audio processing, telecommunications, etc. The time complexity of the classical Fast Fourier Transform (FFT) algorithm proposed by Cooley and Turkey Cooley & Tukey (1965) is $O(n \log n)$, where $n$ is the number of input points. Optics imaging (Voelz, 2011; Goodman, 2017), magnetic resonance image (MRI) (Aibinu et al., 2008) and the physics (Reynolds, 1989) are benefits from this algorithm. The algorithm proposed by Cooley and Turkey Cooley & Tukey (1965) takes $O(n)$ samples to compute the Fourier transformation result. The number of samples taken is an important factor. For example, it influences the amount of ionizing radiation that a patient is exposed to during CT scans. The time that a patient spends within the scanner can also be reduced by taking fewer samples. Thus, we consider the Fourier Transform problems in two computational aspects. Thus, we consider two aspects of the Fourier Transform problems. The first aspect is the reconstruction time which is the time of decoding the signal from the measurements. The second aspect is the sample complexity. Sample complexity is the number of noisy samples required by the algorithm. There is a long line of works optimizing the time and the sample complexity of Fourier Transform in the field of signal-processing and the field of TCS (Cooley & Tukey, 1965; Reynolds, 1989; Aibinu et al., 2008; Voelz, 2011; Hassanieh et al., 2012a; Boashash, 2015).

As a result, we can anticipate that algorithms that leverage sparsity assumptions about the input and outperform FFT in applications will be of significant practical utility. In general, the two most significant factors to optimize are the sample complexity and the time complexity of obtaining the Fourier Transform result. In many real-world applications, computing the approximate Fourier transformation results for a set of selective coordinates is sufficient, and we can leverage the approximation guarantee to accelerate the computation. The set query was originally proposed by Price (2011). The original definition doesn't have restrictions on Fourier measurements. Then, Kapralov (2017) generalizes the classical set query definition (Price, 2011) into the Fourier setting.

In this paper, we consider the set estimation based on the Fourier measurement problem (defined by Kapralov (2017)) where given a vector $x \in \mathbb{C}^n$, approximation parameters $\epsilon, \delta \in (0, 1)$ and a query set $S \subset [n]$ with $|S| = k$, we want to compute an approximate Fourier transform result $x' \in \mathbb{C}^n$ in sublinear time and sample complexity and compared with the standard Fourier transform result $\widehat{x} \in \mathbb{C}^n$, the following approximation guarantee holds:

$$\|(x' - \widehat{x})_S\|_2^2 \leq \epsilon \|\widehat{x}_{\overline{S}}\|_2^2 + \delta \|\widehat{x}\|_1^2$$

with probability at least $9/10$. For a set $S \subset [n]$ and a vector $x \in \mathbb{R}^n$, we define $x_S$ by setting if $i \in S$, $(x_S)_i = x_i$ and otherwise $(x_S)_i = 0$. $\overline{S}$ denotes the complement of the set $S$, i.e., $\overline{S} := [n] \setminus S$.

Table 1: The comparison between our result and the results from prior works (Hassanieh et al., 2012a; Kapralov, 2017).

| References | Samples | Time |
|---|---|---|
| (Hassanieh et al., 2012a) | $\epsilon^{-1}k\log^2(n)$ | $\epsilon^{-1}k\log^2(n)$ |
| (Kapralov, 2017) | $\epsilon^{-1}k$ | $\epsilon^{-1}k\log^{2.1}(n)\log(R^*)$ |
| Ours | $\epsilon^{-1}k\log(n)$ | $\epsilon^{-1}k\log(n)$ |

For this Fourier set query problem, there are two major prior works Kapralov (2017) and Hassanieh et al. (2012a). Kapralov (2017) studies the problem explicitly, whereas Hassanieh et al. (2012a) implicitly provides a solution to the Fourier set query, we will provide more details in the later paragraphs. The work by Kapralov (2017) first explicitly defines the Fourier set query problem and studies it. Kapralov (2017) obtains an algorithm that has sample complexity $O(k/\epsilon)$ and running time $O(\epsilon^{-1}k\log^{2.1}(n)\log(R^*))$ for $\ell_2/\ell_2$ Fourier set query. Here, $R^*$ is an upper bound on the $\|\cdot\|_\infty$ norm of the vector. In most applications, $R^*$ are considered $\mathrm{poly}(n)$. Our approach gives an algorithm with $O(\epsilon^{-1}k\log(n))$ running time. The running time of our result has no dependence on $\log R^*$, but our result does not achieve the optimal sample complexity. Hassanieh et al. (2012a) didn't study the Fourier set query problem, instead, they studied the Fourier sparse recovery problem. However, applying their algorithm from Hassanieh et al. (2012a) to Fourier set query, it provides an algorithm with time complexity of $O(\epsilon^{-1}k\log^2(n))$ and sample complexity of $O(\epsilon^{-1}k\log^2(n))$. Our main contributions are summarized as follows:

- We present an efficient algorithm for the Fourier set query problem.
- We provide comprehensive theoretical guarantees to show the predominance of our algorithms over the existing algorithm.

**Roadmap.** We first present the related work about discrete Fourier transform, continuous Fourier transform and some applications of Fourier transform in Section 2. We define our problem and present our main theorem in Section 3. We present a high-level overview of our techniques in Section 4. We provide some definitions, notations, and technique tools in Section 5. As our main result in this paper, our algorithm (see Algorithm 1) and the analysis of the correctness and complexity of it is given in Section 6. Finally, we conclude our paper in Section 7.

## 2 RELATED WORK

**Discrete Fourier Transform** For computational jobs, among the most crucial and often employed algorithms is the discrete Fourier transform (DFT). There is a long line of works focusing on sparse discrete Fourier transforms. Results can be divided into two kinds: the first kind of results choose sublinear measurements and achieve sublinear or linear recovery time. This kind of work includes Gilbert et al. (2005); Hassanieh et al. (2012a;b); Iwen (2013); Indyk et al. (2014); Indyk & Kapralov (2014); Kapralov (2016; 2017); Nakos et al. (2019).

The second kind of results randomly choose measurements and prove that a generic recovery algorithm succeeds with high probability. A common generic recovery algorithm that this kind of work uses is $\ell_1$ minimization. These results prove the Restricted Isometry Property (Candes et al., 2006; Rudelson & Vershynin, 2008; Bourgain, 2014). Currently, the first kind of solutions have better theoretical guarantees in sample and time complexity. However, the second kind of algorithm has high success probabilities and higher capability in practice.

**Continuous Fourier Transform** Shi et al. (2013) studies sparse Fourier transforms on continuous signals. They apply a discrete sparse Fourier transform algorithm, followed by a hill-climbing method to optimize their solution into a reasonable range. Price & Song (2015) presents an algorithm whose sample complexity is only linear to $k$ and logarithmic in the signal-to-noise ratio. Their frequency resolution is suitable for robustly computing sparse continuous Fourier transforms. Jin et al. (2020) generalizes Price & Song (2015) into high-dimensional setting. Chen et al. (2016) provides an algorithm that supports the reconstruction of a signal without a frequency gap. They present a solution to approximate the signal using a constant factor noise growth and take samples

polynomial in $k$ and logarithmic in the signal-to-noise ratio. Recently Song et al. (2022) improves the approximation ratio of Chen et al. (2016).

**Application of Fourier Transform** Solving partial differential equations is one of the most important applications of Fourier transformation. Some differential equations are simpler to understand in the frequency domain because the action of differentiation in the time domain corresponds to the multiplication by the frequency. Additionally, frequency-domain multiplication is equivalent to convolution in the time domain (McGillem & Cooper, 1991; Proakis, 2001; Friedlander et al., 1998).

Various applications of the Fourier transform include nuclear magnetic resonance (NMR) (Hoult & Bhakar, 1997; Rabi et al., 1938; Schmidt-Rohr & Spiess, 2012), and other types of spectroscopy, such as infrared (FTIR) (Griffiths, 1983). In NMR, a free induction decay (FID) signal with an exponential shape is recorded in the time domain and Fourier transformed into a Lorentzian line-shape in the frequency domain. Mass spectrometry and magnetic resonance imaging (MRI) both employ the Fourier transform. The Fourier transform is also used in quantum mechanics (Wilde, 2013).

For the spectrum analysis of time-series (Schreier & Scharf, 2010; Scharf & Demeure, 1991), the Fourier transform is employed. The Fourier transformation is often not applied to the signal itself in the context of statistical signal processing. It has been discovered in practice that it is best to simulate a signal by a function (or, alternatively, a stochastic process) that is stationary in the sense that its distinctive qualities are constant across all time, even though a genuine signal is in fact transitory. It has been discovered that taking the Fourier transform of the function's autocorrelation function is more advantageous for the analysis of signals since the Fourier transform of such a function does not exist in the conventional sense.

## 3 FOURIER SET QUERY

In Section 3.1, we formally define the problem we study. In Section 3.2, we present our main result.

**Notation** We use $\mathbf{i}$ to denote $\sqrt{-1}$. Note that $e^{\mathbf{i}\theta} = \cos(\theta) + \mathbf{i}\sin(\theta)$. For any complex number $z \in \mathbb{C}$, we have $z = a + \mathbf{i}b$, where $a, b \in \mathbb{R}$. We define the complement of $z \in \mathbb{C}$ as $\overline{z} = a - \mathbf{i}b$, and for a set $S$, we use $\overline{S}$ to denote its complement. We define $|z| := \sqrt{z\overline{z}} = \sqrt{a^2 + b^2}$, and for a set $S$, we use $|S|$ to denote its cardinality. For any complex vector $x \in \mathbb{C}^n$, we use $\mathrm{supp}(x)$ to denote the support of $x$ and define $\|x\|_0 := |\mathrm{supp}(x)|$. We define $\omega = e^{2\pi \mathbf{i}/n}$, which is the $n$-th unitary root i.e. $\omega^n = 1$.

The discrete convolution of functions $f$ and $g$ is given by $(f * g)[n] = \sum_{m=-\infty}^{+\infty} f[m]g[n-m]$. For a complex vector $x \in \mathbb{C}^n$, we use $\widehat{x} \in \mathbb{C}^n$ to denote its Fourier spectrum which is defined as:

$$\widehat{x}_i = \frac{1}{\sqrt{n}} \sum_{j=1}^{n} e^{-2\pi \mathbf{i}ij/n} x_j, \forall i \in [n].$$

Then, the inverse transform is as follows:

$$x_j = \frac{1}{\sqrt{n}} \sum_{i=1}^{n} e^{2\pi \mathbf{i}ij/n} \widehat{x}_i, \forall j \in [n].$$

We define:

$$\mathrm{Err}(x, k) := \min_{k\text{-sparse } y} \|x - y\|_2.$$

We define $x_S$ as a vector by setting if $i \in S$, $(x_S)_i = x_i$ and otherwise $(x_S)_i = 0$, for a vector $x \in \mathbb{R}^n$ and a set $S \subseteq [n]$.

### 3.1 FOURIER SET QUERY PROBLEM

In this section, we give a formal definition of the main problem focused on.

**Definition 3.1** (Sample Complexity). *Given a vector $x \in \mathbb{C}^n$, we say the sample complexity of an algorithm is $c$ (an Algorithm takes $c$ samples) when $c$ is the number of the coordinates used and $c \leq n$.*

**Definition 3.2** (Main problem). *Given a vector $x \in \mathbb{C}^n$, we let $\widehat{x} \in \mathbb{C}^n$ be the Fourier transformation result. For every $\epsilon, \delta \in (0, 1)$, $k \geq 1$, and $S \subseteq [n]$ with $|S| = k$, our goal is to design an algorithm that*

- **Part 1.** *takes samples from $x \in \mathbb{C}^n$ (note that we treat each entry of $x$ as a sample), and*

- **Part 2.** *takes some time to output a vector $x' \in \mathbb{C}^n$ satisfying:*

$$\|(x' - \widehat{x})_S\|_2^2 \leq \epsilon \|\widehat{x}_{\overline{S}}\|_2^2 + \delta \|\widehat{x}\|_1^2.$$

*We want to optimize both sample complexity (which is the number of coordinates we need to access in $x$) and the running time.*

### 3.2 OUR RESULT

Based on the discussion above our main result is presented as follows:

**Theorem 3.3** (Informal version of Theorem 6.3). *Let $x \in \mathbb{C}^n$, $\epsilon \in (0, 1)$, and $\delta \in (0, 1)$. Let $S \subseteq [n]$ satisfying $|S| = k$. Let $\widehat{x} \in \mathbb{C}^n$ be the Fourier transformation result. Then, there exists an algorithm (Algorithm 1) that takes $O(\epsilon^{-1} k \log(n/\delta))$ samples from $x$, runs in $O(\epsilon^{-1} k \log(n/\delta))$ time, and outputs a vector $x' \in \mathbb{C}^n$ satisfying*

$$\|(x' - \widehat{x})_S\|_2^2 \leq \epsilon \|\widehat{x}_{\overline{S}}\|_2^2 + \delta \|\widehat{x}\|_1^2,$$

*with probability at least $9/10$.*

## 4 TECHNIQUE OVERVIEW

In this section, we give an overview of the technique methods used to prove our main result and the complexity analysis about time and sample (see Definition 3.1). First, we give an introduction about the main functions and their time complexity as well as other properties used in our algorithm. Then, based on the functions, we give an analysis of the correctness of our algorithm where, with probability at least $9/10$, it can finally produce a $x'$ which satisfies

$$\|(x' - \widehat{x})_S\|_2^2 \leq \epsilon \|\widehat{x}_{\overline{S}}\|_2^2 + \delta \|\widehat{x}\|_1^2.$$

The analysis of total complexity comes last, with $O(\epsilon^{-1} k \log(n/\delta))$ as the sample complexity (see Definition 3.1) and $O(\epsilon^{-1} k \log(n/\delta))$ as the time complexity. And then we can make sure the algorithm solves the problem (see Definition 3.2) with better performance compared to the prior works Kapralov (2017) and Hassanieh et al. (2012a) (see details in Table 1).

**Technique I: HASHTOBINS** We first give an overview of the techniques we use from Hassanieh et al. (2012a). We use the same function HASHTOBINS with the one in Hassanieh et al. (2012a), which is one of the key parts of the function EstimateValues. We can attain a $\widehat{u}$, where the $\widehat{u}_j$ for satisfies the following equation

$$\widehat{u}_j = \sum_{h_{\sigma, b}(i) = j} \widehat{(x - z)}_i (\widehat{G'_{B, \delta, \alpha}})_{-o_{\sigma, b}(i)} \omega^{a \sigma i} \pm \delta \|\widehat{x}\|_1,$$

where $\widehat{G'_{B, \delta, \alpha}} \in \mathbb{R}^n$ is defined as in Definition 5.3 when we formally present the techniques from prior works (also see Figure 1 for a more explicit visualization). $h_{\sigma, b}(i)$ and $o_{\sigma, b}(i)$ are the hash and the offset function, respectively. We use the hashing scheme and filtering to isolate frequencies (see the concept of well-isolate we develop in the next paragraph). To help the analysis of the time complexity of our Algorithm 1, the time complexity of the function above is $O(\frac{B}{\alpha} \log(n/\delta) + \|\widehat{z}\|_0 + \zeta \log(n/\delta))$, with $\zeta = |\{i \in \text{supp}(\widehat{z}) \mid E_{\text{off}}(i)\}|$.

**Technique II: Query Set $S$** Now, we summarize the techniques we develop to support our main result. We use $S$ as the query set and $S_i$ is the set attained by updating $S$ with $i - 1$ iterations. And we use $k_i = k \gamma^{i-1}$ where $\gamma \leq \frac{1}{1000}$ and $k \geq 1$. We demonstrate that we can successfully complete our query, i.e., we can compress $S_i$ to a small enough size such that $|S_i| \leq k_i$.

Given a vector $x$ and $t \in [n]$ as a coordinate of it, we also define "well-isolated" based on the concepts above, which are frequencies that don't suffer from significant interference from other frequencies in the current iteration. Then, we can prove that with probability at least $1 - 6\alpha_i$, where $\alpha_i = 1/(200i^3)$, we have that $t$ is "well-isolated". From the definition of well-isolated (see Definition 5.10), it suffices to bound the probabilities of large offset, large noise, and collision (see details of the proof in Appendix B.1). This is a crucial property that we frequently use when showing correctness and complexity: we show our algorithm can estimate these well-isolated frequencies accurately. By setting $|S_i| \leq k_i$ and performing a sufficiently large number of iterations, we can ensure that $|S_i|$ (the number of unfinished queries) is sufficiently small.

**Technique III: Correctness and Complexity** By the upper bound of $\|\widehat{x}_{\overline{S}_{i+1}}^{(i+1)}\|_2^2$, we can obtain the upper bound of error. With probability $1 - 10\alpha_i/\gamma$, we can have

$$\|\widehat{x}_{\overline{S}_{i+1}}^{(i+1)}\|_2^2 \leq (1 + \epsilon_i)\|\widehat{x}_{\overline{S}_i}^{(i)}\|_2^2 + \epsilon_i\delta^2 n\|\widehat{x}\|_1^2,$$

through combining the concepts of "well-isolated" coordinates (see Definition 5.10), probabilistic inequalities, and inductive argument (see Appendix B.2 for details).

Then we can demonstrate:

$$\|\widehat{x}_S - \widehat{z}^{(R+1)}\|_2^2 \leq \epsilon(\|\widehat{x}_{\overline{S}}\|_2^2 + \delta^2 n\|\widehat{x}\|_1^2). \tag{1}$$

The proof also leverages the iterative nature of the algorithm, which runs for $R + 1$ iterations (where $R = \log k$). Each iteration improves the estimate of the Fourier coefficients. To bound the error in each iteration, we combine the property of well-isolated coordinates and the properties of the HashToBins function that we derive in Lemma B.3, namely

$$\Pr\left[\left\|\widehat{x}_{T_i}^{(i)} - \widehat{w}^{(i)}\right\|_2^2 \leq \frac{\epsilon_i}{20}(\|\widehat{x}_{\overline{S}_i}^{(i)}\|_2^2 + \delta^2 n\|\widehat{x}\|_1^2)\right] \geq 1 - \alpha_i.$$

Notice that the $\widehat{z}^{(R+1)}$ in Eq. (1) is the output of our Algorithm 1 which is also the $x'$ in our problem (see Definition 3.2). The above inequalities demonstrate that the Algorithm 1 constructed by us can output a $x'$ which satisfies

$$\|(\widehat{x} - x')_S\|_2^2 \leq \epsilon\|\widehat{x}_{\overline{S}}\|_2^2 + \delta\|\widehat{x}\|_1^2$$

with a success probability of 9/10. We obtain the sample complexity and time complexity as follows:

$$\sum_{i=1}^{R}(B_i/\alpha_i)\log(n/\delta) = \epsilon^{-1}k\log(n/\delta),$$

where $B_i \geq 1000k_i/(\alpha_i^2\epsilon_i)$.

## 5 PRELIMINARY

In Section 5.1, we present the basic concepts related to the Fourier transform. We introduce some technical tools in Section 5.2. Then we introduce spectrum permutations and filter functions in Section 5.3. They are used as hashing schemes in the Fourier transform literature. In Section 5.4, we introduce collision events, large offset events, and large noise events.

### 5.1 FOURIER TRANSFORM

The discrete convolution of functions $f$ and $g$ is defined as follow

$$(f * g)[n] := \sum_{m=-\infty}^{+\infty} f[m]g[n - m]$$

Let $x \in \mathbb{C}^n$ be a complex vector. The Fourier spectrum of $x$ is denoted by $\widehat{x} \in \mathbb{C}^n$

$$\widehat{x}_i := \frac{1}{\sqrt{n}}\sum_{j=1}^{n} e^{-2\pi \mathbf{i}ij/n}x_j, \forall i \in [n].$$

Then, the inverse transform can be obtained $x_j = \frac{1}{\sqrt{n}}\sum_{i=1}^{n} e^{2\pi \mathbf{i}ij/n}\widehat{x}_i, \forall j \in [n]$. We define

$$\mathrm{Err}(x, k) := \min_{k\text{-sparse } y}\|x - y\|_2.$$

## 5.2 Technical Tools

We show several technical tools and some lemmas in prior works we used in the following section.

**Lemma 5.1** (Markov's inequality). *If $X$ is a nonnegative random variable and $a > 0$, then the probability that $X$ is at least $a$ is at most the expectation of $X$ divided by $a$: $\Pr[X \geq a] \leq \frac{\mathbb{E}(X)}{a}$.*

*Let $a = \widetilde{a} \cdot \mathbb{E}(X)$ (where $\widetilde{a} > 0$). Then we can rewrite the previous inequality as $\Pr[X \geq \widetilde{a} \cdot \mathbb{E}(X)] \leq \frac{1}{\widetilde{a}}$*

The following two lemmas of complex numbers are standard. We prove the following two lemmas for the completeness of the paper (see Section A.2).

**Lemma 5.2.** *Given a fixed vector $x \in \mathbb{R}^n$ and a pairwise independent random variable $\sigma_i$ where $\sigma_i = \pm 1$ with probability $1/2$ respectively. Then we have: $\mathbb{E}_\sigma[(\sum_{i=1}^n \sigma_i x_i)^2] = \|x\|_2^2$*

**Lemma 5.3.** *Let $a \sim [n]$ uniformly at random. Given a fixed vector $x \in \mathbb{C}^n$ and $\omega^{\sigma a i}$, then we have:*

$$\mathbb{E}_a[|\sum_{i=1}^n x_i \omega^{\sigma a i}|^2] = \|x\|_2^2$$

## 5.3 Permutation and filter function

We use the same (pseudorandom) spectrum permutation as Hassanieh et al. (2012a):

**Definition 5.4.** *Let $a$ and $b$ are positive integers in $[n]$. Suppose $\sigma^{-1}$ exists $\pmod{n}$. We define the permutation $P_{\sigma,a,b}$ by*

$$(P_{\sigma,a,b}x)_i := x_{\sigma(i-a)} e^{-2\pi \mathbf{i}\sigma b i/n},$$

*where $\mathbf{i} = \sqrt{-1}$ and $i$ denote the $i$-th entry of $P_{\sigma,a,b}x$. We also define $\pi_{\sigma,b} := \sigma(i - b) \pmod{n}$.*

**Claim 5.5** (Claim 2.2 in Hassanieh et al. (2012a)). *Let $P_{\sigma,a,b}x$ be defined as in Definition 5.4. We have that*

$$\widehat{P_{\sigma,a,b}x}_{\pi_{\sigma,b}(i)} = \widehat{x}_i e^{-2\pi \mathbf{i}\sigma a i/n}.$$

$h_{\sigma,b}(i)$ is defined as the "bin" with the mapping of frequency $i$ onto. We define $o_{\sigma,b}(i)$ as the "offset". We formally define them as follows:

**Definition 5.6.** *Let the hash function be defined as $h_{\sigma,b}(i) := \text{round}(\frac{\pi_{\sigma,b}(i)B}{n})$.*

**Definition 5.7.** *Let the offset function be defined as $o_{\sigma,b}(i) := \pi_{\sigma,b}(i) - h_{\sigma,b}(i)\frac{n}{B}$.*

In this paper, we use the same filter function as Hassanieh et al. (2012a); Price & Song (2015); Chen et al. (2016):

**Definition 5.8.** *Given parameters $B \geq 1$, $\delta > 0$, $\alpha > 0$, we say that $(G, \widehat{G}') = (G_{B,\delta,\alpha}, \widehat{G}'_{B,\delta,\alpha}) \in \mathbb{R}^n \times \mathbb{R}^n$ is a filter function if it satisfies the following properties:*

1. $|\text{supp}(G)| = O(\alpha^{-1}B\log(n/\delta))$.

2. *if $|i| \leq (1-\alpha)n/(2B)$, $\widehat{G}'_i = 1$.*

3. *if $|i| \geq n/(2B)$, $\widehat{G}'_i = 0$.*

4. *for all $i$, $\widehat{G}'_i \in [0,1]$.*

5. $\left\|\widehat{G}' - \widehat{G}\right\|_\infty < \delta$.

## 5.4 Collision event, large offset event, and large noise event

We use three types of events defined in Hassanieh et al. (2012a) as basic building blocks for analyzing Fourier set query algorithms. For any $i \in S$, we define three types of events associated with $i$ and $S$ and defined over the probability space induced by $\sigma$ and $b$:

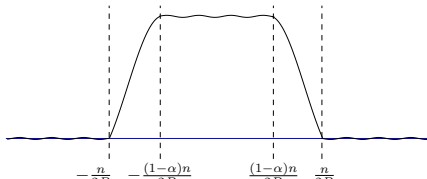

$$-\frac{n}{2B} \qquad -\frac{(1-\alpha)n}{2B} \qquad \frac{(1-\alpha)n}{2B} \qquad \frac{n}{2B}$$

Figure 1: Filter $\widehat{G}'$

**Definition 5.9** (Collision, large offset, large noise). *The definition of three events are given as follow:*

- *We say "Large offset" event $E_{\mathrm{off}}(i)$ holds if $n(1-\alpha)/(2B) \leq |o_{\sigma,b}(i)|$.*

- *We say "Large noise" event $E_{\mathrm{noise}}(i)$ holds if*

$$(\alpha B)^{-1} \cdot \mathrm{Err}^2(\widehat{x}', k) \leq \mathbb{E}\left[\left\|\widehat{x}'_{h_{\sigma,b}^{-1}(h_{\sigma,b}(i))\setminus S}\right\|_2^2\right].$$

- *We say "Collision" event $E_{\mathrm{coll}}(i)$ holds if $h_{\sigma,b}(i) \in h_{\sigma,b}(S\setminus\{i\})$.*

**Definition 5.10** (Well-isolated). *For a vector $x \in \mathbb{R}^n$, we say a coordinate $t \in [n]$ is "well isolated" when none of "Collision" event, "Large offset" and "Large noise" event holds.*

## 6 ANALYSIS ON FOURIER SET QUERY ALGORITHM

In this section, we will give an total analysis about our Algorithm 1. First, we provide the iterative loop analysis which is the main part of our main function FOURIERSETQUERY in Section 6.1. By this analysis, we demonstrate an important property of the Algorithm 1 in Section 6.2. In Section 6.3, we prove the correctness of the algorithm. We also provide the analysis of the complexity (sample and time) of Algorithm 1. Then we can give a satisfying answer to the problem (see Definition 3.2) with Algorithm 1 attained by us whose performance (on sample and time complexity) is better than prior works (see Table 1).

### 6.1 ITERATIVE LOOP ANALYSIS

Iterative loop analysis for Fourier set query is more tricky than the classic set query, because in the Fourier case, hashing is not perfect, in the sense that by using spectrum permutation and filter function (as the counterpart of hashing techniques), one coordinate can non-trivially contribute to multiple bins.

**Lemma 6.1** (Informal version of Lemma B.1). *Consider an arbitrary filtering step $i$. Let $x \in \mathbb{R}^n$, $\gamma \leq 1/1000$, $\alpha_i = 1/(200i^3)$, for a coordinate $t \in [n]$ and each $i \in [R]$, with probability at least $1 - 6\alpha_i$. Then, $t$ is "well isolated" (see Definition 5.10).*

**Lemma 6.2** (Informal version of Lemma B.2). *Let $C \geq 1000$ and $\gamma \leq 1/1000$. For all $k \geq 1$ and $\epsilon_i \in (0, 1)$, we define*

$$k_i := k\gamma^{i-1},$$
$$\epsilon_i := \epsilon(10\gamma)^i,$$
$$\alpha_i := 1/(200i^3),$$
$$B_i := C \cdot k_i/(\alpha_i^2 \epsilon_i).$$

*Let $R > 1$. If for all $i \in [R]$, for all $j \in [i-1]$, we have*

1. $\mathrm{supp}(\widehat{w}^{(j)}) \subseteq S_j$.

2. $|S_{j+1}| \leq k_{j+1}$.

3. $\widehat{z}^{(j+1)} = \widehat{z}^{(j)} + \widehat{w}^{(j)}$.

**Algorithm 1** Fourier set query algorithm

1: **procedure** FOURIERSETQUERY($x, S, \epsilon, k$) ▷ Theorem 6.5
2:     $\gamma \leftarrow 1/1000, C \leftarrow 1000, \widehat{z}^{(1)} \leftarrow 0, S_1 \leftarrow S$
3:     **for** $i = 1 \rightarrow R$ **do**
4:         $k_i \leftarrow k\gamma^i, \epsilon_i \leftarrow \epsilon(10\gamma)^i, \alpha_i \leftarrow 1/(100i^3), B_i \leftarrow C \cdot k_i/(\alpha_i^2 \epsilon_i)$
5:         $\widehat{w}^{(i)}, T_i \leftarrow$ ESTIMATEVALUES($x, \widehat{z}^{(i)}, S_i, B_i, \delta, \alpha_i$) ▷ $\widehat{w}^{(i)}$ is $|T_i|$-sparse
6:         $S_{i+1} \leftarrow S_i \backslash T_i$
7:         $\widehat{z}^{(i+1)} \leftarrow \widehat{z}^{(i)} + \widehat{w}^{(i)}$
8:     **end for**
9:     **return** $\widehat{z}^{(R+1)}$
10: **end procedure**
11: **procedure** ESTIMATEVALUES($x, \widehat{z}, S, B, \delta, \alpha$) ▷ Lemma 6.3
12:     Choose $a, b \in [n]$ uniformly at random
13:     Choose $\sigma$ uniformly at random from the set of odd numbers in $[n]$
14:     $\widehat{u} \leftarrow$ HASHTOBINS($P_{\sigma,a,b}, \alpha, \widehat{z}, B, \delta, x$)
15:     $\widehat{w} \leftarrow 0, T \leftarrow \emptyset$
16:     **for** $t \in S$ **do**
17:         **if** $t$ is isolated from other coordinates of $S$ **then** ▷ $h_{\sigma,b}(t) \notin h_{\sigma,b}(S \backslash \{t\})$
18:             **if** no large offset **then** ▷ $n(1-\alpha)/(2B) > |o_{\sigma,b}(t)|$
19:                 $\widehat{w}_t \leftarrow \widehat{u}_{h_{\sigma,b}(t)} e^{-\frac{2\pi \mathbf{i}}{n}\sigma a t}$
20:                 $T \leftarrow T \cup \{t\}$
21:             **end if**
22:         **end if**
23:     **end for**
24:     **return** $\widehat{w}, T$
25: **end procedure**
26: **procedure** HASHTOBINS($P_{\sigma,a,b}, \alpha, \widehat{z}, B, \delta, x$)
27:     Compute $\widehat{y}_{jn/B}$ for $j \in [B]$, where $y = G_{B,\alpha,\delta} \cdot (P_{\sigma,a,b}x)$
28:     Compute $\widehat{y}'_{jn/B} = \widehat{y}_{jn/B} - (\widehat{G'_{B,\alpha,\delta}} * \widehat{P_{\sigma,a,b}z})_{jn/B}$
29:     **return** $\widehat{u}_j = \widehat{y}'_{jn/B}$
30: **end procedure**

4. $\widehat{x}^{(j+1)} = \widehat{x} - \widehat{z}^{(j+1)}$.

5. $\|\widehat{x}^{(j+1)}_{\overline{S}_{j+1}}\|_2^2 \leq (1 + \epsilon_j)\|\widehat{x}^{(j)}_{\overline{S}_j}\|_2^2 + \epsilon_j \delta^2 n \|\widehat{x}\|_1^2$,

then, with probability $1 - 10\alpha_i/\gamma$, we have $|S_{i+1}| \leq k_{i+1}$.

**Lemma 6.3** (Informal version of Lemma B.4). *Let $C \geq 1000$ and $\gamma \leq 1/1000$. If for all $k \geq 1$, $\epsilon \in (0, 1)$, $R \geq 1$, $i \in [R]$, and $j \in [i-1]$, we have*

1. supp($\widehat{w}^{(j)}$) $\subseteq S_j$.

2. $|S_{j+1}| \leq k_{j+1}$.

3. $\widehat{z}^{(j+1)} = \widehat{z}^{(j)} + \widehat{w}^{(j)}$.

4. $\widehat{x}^{(j+1)} = \widehat{x} - \widehat{z}^{(j+1)}$.

5. $\|\widehat{x}^{(j+1)}_{\overline{S}_{j+1}}\|_2^2 \leq (1 + \epsilon_j)\|\widehat{x}^{(j)}_{\overline{S}_j}\|_2^2 + \epsilon_j \delta^2 n \|\widehat{x}\|_1^2$.

*Then, with probability $1 - 10\alpha_i/\gamma$, we have*

1. supp($\widehat{w}^{(i)}$) $\subseteq S_i$.

2. $|S_{i+1}| \leq k_{i+1}$.

3. $\widehat{z}^{(i+1)} = \widehat{z}^{(i)} + \widehat{w}^{(i)}$.

4. $\widehat{x}^{(i+1)} = \widehat{x} - \widehat{z}^{(i+1)}$.

5. $\|\widehat{x}^{(i+1)}_{\overline{S}_{i+1}}\|_2^2 \leq (1 + \epsilon_i)\|\widehat{x}^{(i)}_{\overline{S}_i}\|_2^2 + \epsilon_i \delta^2 n \|\widehat{x}\|_1^2$.

The proofs are deferred to Section B.1.

## 6.2 INDUCTION TO ALL THE ITERATIONS

For completeness, we give the induced result among all the iterations ($i \in [R]$). By the following lemma at hand, we can finally attain the theorem in Section 6.3.

**Lemma 6.4.** *Given parameters $C \geq 1000$, $\gamma \leq 1/1000$. For any $k \geq 1$, $\epsilon \in (0, 1)$, $R \geq 1$. For each $i \in [R]$, we have with probability $1 - 10\alpha_i/\gamma$, we have*

$$|S_{i+1}| \leq k_i$$

*and*

$$\|\widehat{x}^{(i+1)}_{\overline{S}_{i+1}}\|_2^2 \leq (1 + \epsilon_i)\|\widehat{x}^{(i)}_{\overline{S}_i}\|_2^2 + \epsilon_i \delta^2 n \|\widehat{x}\|_1^2$$

The details of the proof are provided in Section B.2.

## 6.3 MAIN RESULT

In this subsection, we give the main result as the following theorem.

**Theorem 6.5** (Main result, formal version of Theorem 3.3)**.** *If all of the conditions are met*

- **Condition 1.** *Let $x \in \mathbb{C}^n$, $\epsilon \in (0, 1)$, $\delta \in (0, 1)$.*

- **Condition 2.** *We denote $\widehat{x}$ as the Fourier transformation result.*

- **Condition 3.** *We define $S \subset [n]$, $|S| = k$ where $k \geq 1$.*

*An algorithm (Algorithm 1) exists such that*

- **Part 1.** *It takes $O(\epsilon^{-1}k \log(n/\delta))$ samples from $x$.*

- **Part 2.** *It runs in $O(\epsilon^{-1}k \log(n/\delta))$.*

- **Part 3.** *It holds with probability at least $9/10$.*

- **Part 4.** *It outputs a vector $x' \in \mathbb{C}^n$ such that*

$$\|(x' - \widehat{x})_S\|_2^2 \leq \epsilon\|\widehat{x}_{\overline{S}}\|_2^2 + \delta\|\widehat{x}\|_1^2,$$

The proof is deferred to Section B.3.

## 7 CONCLUSION

Fourier transformation is an intensively researched topic in a variety of scientific disciplines. Numerous applications exist within machine learning, signal processing, compressed sensing, etc. In this paper, we study the problem of Fourier set query. With an approximation parameter $\epsilon$, a vector $x \in \mathbb{C}^n$ and a query set $S \subset [n]$ of size $k$, our algorithm uses $O(\epsilon^{-1}k \log(n/\delta))$ Fourier measurements, runs in $O(\epsilon^{-1}k \log(n/\delta))$ time and outputs a vector $x'$ such that $\|(x' - \widehat{x})_S\|_2^2 \leq \epsilon\|\widehat{x}_{\overline{S}}\|_2^2 + \delta\|\widehat{x}\|_1^2$ with probability of at least $9/10$. Currently, our result only holds for $\ell_2$, generalizing results to $\ell_p$ norm could be an interesting future direction. This work is purely a theoretical result, we don't know any negative societal impact.

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

APPENDIX

**Roadmap.** In Section A, we introduce notations and technique tools used in our paper. In Section B, our analysis of our main result is presented.

## A  PRELIMINARY

In this section, we first present some definitions and background for Fourier transform in Section A.1. We introduce some technical tools in Section A.2. Then we introduce spectrum permutations and filter functions in Section A.3. They are used as hashing schemes in the Fourier transform literature. In Section A.4, we introduce collision events. large offset events, and large noise events.

### A.1  NOTATIONS

We use $\mathbf{i}$ to denote $\sqrt{-1}$. Note that $e^{\mathbf{i}\theta} = \cos(\theta) + \mathbf{i}\sin(\theta)$. For any complex number $z \in \mathbb{C}$, we have $z = a + \mathbf{i}b$, where $a, b \in \mathbb{R}$. We define the complement of $z$ as $\overline{z} = a - \mathbf{i}b$. We define $|z| = \sqrt{z\overline{z}} = \sqrt{a^2 + b^2}$. For any complex vector $x \in \mathbb{C}^n$, we use $\mathrm{supp}(x)$ to denote the support of $x$, and then $\|x\|_0 = |\mathrm{supp}(x)|$. We define $\omega = e^{2\pi\mathbf{i}/n}$, which is the $n$-th unitary root i.e. $\omega^n = 1$.

The discrete convolution of functions $f$ and $g$ is given by,

$$(f * g)[n] = \sum_{m=-\infty}^{+\infty} f[m]g[n-m]$$

For a complex vector $x \in \mathbb{C}^n$, we use $\widehat{x} \in \mathbb{C}^n$ to denote its Fourier spectrum,

$$\widehat{x}_i = \frac{1}{\sqrt{n}} \sum_{j=1}^{n} e^{-2\pi\mathbf{i}ij/n} x_j, \forall i \in [n].$$

Then the inverse transform is

$$x_j = \frac{1}{\sqrt{n}} \sum_{i=1}^{n} e^{2\pi\mathbf{i}ij/n} \widehat{x}_i, \forall j \in [n].$$

We define

$$\mathrm{Err}(x, k) := \min_{k\text{-sparse } y} \|x - y\|_2.$$

We define $x_S$ as a vector by setting if $i \in S$, $(x_S)_i = x_i$ and otherwise $(x_S)_i = 0$, for a vector $x \in \mathbb{R}^n$ and a set $S \subseteq [n]$.

### A.2  TECHNICAL TOOLS

We show several technical tools and some lemmas in prior works we used in the following section.

**Lemma A.1** (Markov's inequality)**.** *If $X$ is a nonnegative random variable and $a > 0$, then the probability that $X$ is at least $a$ is at most the expectation of $X$ divided by $a$:*

$$\Pr[X \geq a] \leq \frac{\mathbb{E}(X)}{a}.$$

*Let $a = \widetilde{a} \cdot \mathbb{E}(X)$ (where $\widetilde{a} > 0$); then we can rewrite the previous inequality as*

$$\Pr[X \geq \widetilde{a} \cdot \mathbb{E}(X)] \leq \frac{1}{\widetilde{a}}$$

The following two lemmas of complex number are standard. We prove the following two lemmas for the completeness of the paper.

**Lemma A.2.** *Given a fixed vector $x \in \mathbb{R}^n$ and a pairwise independent random variable $\sigma_i$ where $\sigma_i = \pm 1$ with probability $1/2$ respectively. Then we have:*

$$\mathbb{E}_{\sigma}[(\sum_{i=1}^{n} \sigma_i x_i)^2] = \|x\|_2^2$$

*Proof.* We have:

$$\mathbb{E}_{\sigma}[(\sum_{i=1}^{n} \sigma_i x_i)^2]$$

$$= \mathbb{E}[\sum_{i=1}^{n} \sigma_i^2 x_i^2] + \mathbb{E}[\sum_{i \neq j} \sigma_i x_i \sigma_j x_j]$$

$$= \mathbb{E}[\sum_{i=1}^{n} \sigma_i^2 x_i^2] + \sum_{i \neq j} \mathbb{E}[\sigma_i \sigma_j] x_i x_j$$

$$= \mathbb{E}[\sum_{i=1}^{n} \sigma_i^2 x_i^2] + \sum_{i \neq j} \mathbb{E}[\sigma_i] \cdot \mathbb{E}[\sigma_j] x_i x_j$$

$$= \mathbb{E}[\sum_{i=1}^{n} \sigma_i^2 x_i^2] + 0$$

$$= \|x\|_2^2$$

where the first step comes from the linearity of expectation, the second step follows the linearity of expectation, the third step $\sigma_i$ is a pairwise independent random variable, the fourth step follows that $\mathbb{E}[\sigma_i] = 0$ , and the final step comes from the definition of $\| \cdot \|_2$ and $\sigma_i^2 = 1$. $\qquad \square$

**Lemma A.3.** *Let $a \sim [n]$ uniformly at random. Given a fixed vector $x \in \mathbb{C}^n$ and $\omega^{\sigma a i}$, then we have:*

$$\mathbb{E}_{a}[|\sum_{i=1}^{n} x_i \omega^{\sigma a i}|^2] = \|x\|_2^2$$

*Proof.* For any fixed $i \in [n]$, we have the inequality as follows

$$\mathbb{E}_{a}[\omega^{ai}] = \frac{1}{n}\sum_{a=1}^{n} \omega^{ai} = \frac{1}{n} \cdot \frac{1 - \omega^{ni}}{1 - \omega^i} = 0 \qquad (2)$$

where the first step comes from geometric sum, and the second step comes from We have:

$$\mathbb{E}_{a}[|\sum_{i=1}^{n} x_i \omega^{\sigma a i}|^2]$$

$$= \mathbb{E}_{a}[(\sum_{i=1}^{n} x_i \omega^{\sigma a i})(\sum_{i=1}^{n} \bar{x}_i \omega^{-\sigma a i})]$$

$$= \mathbb{E}_{a}[\sum_{i=1}^{n} x_i \bar{x}_i] + \mathbb{E}_{a}[\sum_{i \neq j} x_i \omega^{\sigma a i} \bar{x}_j \omega^{-\sigma a j}]$$

$$= \mathbb{E}_{a}[\sum_{i=1}^{n} x_i \bar{x}_i] + \sum_{i \neq j} \mathbb{E}_{a}[\omega^{\sigma a(i-j)}] x_i \bar{x}_j$$

$$= \mathbb{E}_{a}[\sum_{i=1}^{n} x_i \bar{x}_i] + 0$$

$$= \|x\|_2^2$$

where the first step follows that for a complex number $z$, $|z|^2 = z\bar{z}$, the second step follows the linearity of expectation, the third step follows the linearity of expectation, where the fourth step follows Eq.2, and the final step comes from the definition of $\| \cdot \|_2$. $\qquad \square$

## A.3 PERMUTATION AND FILTER FUNCTION

We use the same (pseudorandom) spectrum permutation as Hassanieh et al. (2012a),

**Definition A.4.** *Suppose $\sigma^{-1}$ exists mod $n$. We define the permutation $P_{\sigma,a,b}$ by*

$$(P_{\sigma,a,b}x)_i = x_{\sigma(i-a)}e^{-2\pi \mathbf{i}\sigma bi/n}.$$

We also define $\pi_{\sigma,b} = \sigma(i-b) \pmod n$. Then we have

**Claim A.5.** *We have that*

$$\widehat{P_{\sigma,a,b}x}_{\pi_{\sigma,b}(i)} = \widehat{x}_i e^{-2\pi \mathbf{i}\sigma ai/n}.$$

$h_{\sigma,b}(i)$ is defined as the "bin" with the mapping of frequency $i$ onto. We define $o_{\sigma,b}(i)$ as the "offset". We formally define them as follows:

**Definition A.6.** *Let the hash function be defined as*

$$h_{\sigma,b}(i) := \mathrm{round}(\frac{\pi_{\sigma,b}(i)B}{n}).$$

**Definition A.7.** *Let the offset function be defined as*

$$o_{\sigma,b}(i) := \pi_{\sigma,b}(i) - h_{\sigma,b}(i)\frac{n}{B}.$$

We use the same filter function as Hassanieh et al. (2012a); Price & Song (2015); Chen et al. (2016),

**Definition A.8.** *Given parameters $B \geq 1$, $\delta > 0$, $\alpha > 0$. We say that $(G, \widehat{G}') = (G_{B,\delta,\alpha}, \widehat{G}'_{B,\delta,\alpha}) \in \mathbb{R}^n$ is a filter function if it satisfies the following properties:*

1. *$|\mathrm{supp}(G)| = O(\alpha^{-1}B\log(n/\delta))$.*

2. *if $|i| \leq (1-\alpha)n/(2B)$, $\widehat{G}'_i = 1$.*

3. *if $|i| \geq n/(2B)$, $\widehat{G}'_i = 0$.*

4. *for all $i$, $\widehat{G}'_i \in [0,1]$.*

5. *$\left\|\widehat{G}' - \widehat{G}\right\|_\infty < \infty$.*

## A.4 COLLISION EVENT, LARGE OFFSET EVENT, AND LARGE NOISE EVENT

We use three types of events defined in Hassanieh et al. (2012a) as basic building blocks for analyzing Fourier set query algorithms. For any $i \in S$, we define three types of events associated with $i$ and $S$ and defined over the probability space induced by $\sigma$ and $b$:

**Definition A.9** (Collision, large offset, large noise). *The definition of three events are given as follow:*

- *We say "Large offset" event $E_{\mathrm{off}}(i)$ holds if*

$$n(1-\alpha)/(2B) \leq |o_{\sigma,b}(i)|.$$

- *We say "Large noise" event $E_{\mathrm{noise}}(i)$ holds if*

$$(\alpha B)^{-1} \cdot \mathrm{Err}^2(\widehat{x}', k) \leq \mathbb{E}\left[\left\|\widehat{x}'_{h_{\sigma,b}^{-1}(h_{\sigma,b}(i))\backslash S}\right\|_2^2\right].$$

- *We say "Collision" event $E_{\mathrm{coll}}(i)$ holds if*

$$h_{\sigma,b}(i) \in h_{\sigma,b}(S\backslash\{i\}).$$

**Definition A.10** (Well-isolated). *For a vector $x \in \mathbb{R}^n$, we say a coordinate $t \in [n]$ is "well isolated" when none of "Collision" event, "Large offset" and "Large noise" event holds.*

**Claim A.11** (Claim 3.1 in Hassanieh et al. (2012a)). *For all $i \in S$, we have*

$$\Pr[E_{\text{coll}}(i)] \leq 4\frac{|S|}{B}.$$

**Claim A.12** (Claim 3.2 in Hassanieh et al. (2012a)). *For all $i \in S$, we have*

$$\Pr[E_{\text{off}}(i)] \leq \alpha.$$

**Claim A.13** (Claim 4.1 in Hassanieh et al. (2012a)). *For any $i \in S$, the event $E_{\text{noise}(i)}$ holds with probability at most $4\alpha$*

$$\Pr[E_{\text{noise}(i)}] \leq 4\alpha.$$

**Lemma A.14** (Lemma 4.2 in Hassanieh et al. (2012a)). *With $B$ divide $n$, $a$ uniformly sampled from $[n]$ and the others without limitation in*

$$\widehat{u} = \textsc{HashToBins}(P_{\sigma,a,b}, \alpha, \widehat{z}, B, \delta, x).$$

*With all of $E_{\text{off}}(i)$, $E_{\text{coll}}(i)$ and $E_{\text{noise}}(i)$ not holding and $j = h_{\sigma,b}(i)$, we have for all $i \in [n]$,*

$$\mathbb{E}\left[\left|\widehat{x}'_i e^{-\frac{2\pi \mathbf{i}}{n}a\sigma i}\right|^2 - \widehat{u}_j\right] \leq 2\frac{\rho^2}{\alpha B}.$$

**Lemma A.15** (Lemma 3.3 in Hassanieh et al. (2012a)). *Suppose $B$ divides $n$. The output $\widehat{u}$ of* \textsc{HashToBins} *satisfies*

$$\widehat{u}_j = \sum_{h_{\sigma,b}(i)=j} \widehat{(x-z)}_i (\widehat{G'_{B,\delta,\alpha}})_{-o_{\sigma,b}(i)} \omega^{a\sigma i} \pm \delta\|\widehat{x}\|_1.$$

*Let*

$$\zeta := |\{i \in \text{supp}(\widehat{z}) \mid E_{\text{off}}(i)\}|.$$

*The running time of* \textsc{HashToBins} *is*

$$O(\frac{B}{\alpha}\log(n/\delta) + \|\widehat{z}\|_0 + \zeta\log(n/\delta)).$$

# B ANALYSIS ON FOURIER SET QUERY ALGORITHM

In this section, we will give an total analysis about our Algorithm 1. First, we will provide the iterative loop analysis which is the main part of our main function \textsc{FourierSetQuery} in Section B.1. By this analysis, we demonstrate an important property of the Algorithm 1 in Section B.2. In Section B.3, we prove the the correctness of the algorithm. We also provide the analysis of the complexity (sample and time) of Algorithm 1. Then we can give an satisfying answer to the problem (See Definition 3.2) with Algorithm 1 attained by us whose performance (on sample and time complexity) is better than prior works (See Table 1).

## B.1 ITERATIVE LOOP ANALYSIS

Iterative loop analysis for Fourier set query is more tricky than the classic set query, because in the Fourier case, hashing is not perfect, in the sense that by using spectrum permutation and filter function (as the counterpart of hashing techniques), one coordinate can non-trivially contribute to multiple bins. We give iterative loop induction in Lemma B.4.

**Lemma B.1.** *Given a vector $x \in \mathbb{R}^n$, $\gamma \leq 1/1000$, $\alpha_i = 1/(200i^3)$, for a coordinate $t \in [n]$ and each $i \in [R]$, with probability at least $1 - 6\alpha_i$, We say that $t$ is "well isolated" (See Definition 5.10).*

*Proof.* **Collision.** Using Claim A.11, for any $t \in S_i$, the event $E_{\text{coll}}(t)$ holds with probability at most

$$4|S_i|/B_i \leq \frac{4k_i}{Ck_i/(\alpha_i^2\epsilon_i)}$$
$$= 4\alpha_i^2\epsilon_i/C$$
$$\leq \alpha_i,$$

where the first step follows from the definition of $B_i$ and the assumption on $|S_i|$, the second step is straightforward, the third step follows from the definition of $\epsilon_i$, $\alpha_i$, and $C$.

It means

$$\Pr_{\sigma,b}[E_{\mathrm{coll}}(t)] \leq \alpha_i.$$

**Large offset.** Using Claim A.12, for any $t \in S_i$, the event $E_{\mathrm{off}}(t)$ holds with probability at most $\alpha_i$, i.e.

$$\Pr_{\sigma,b}[E_{\mathrm{off}}(t)] \leq \alpha_i.$$

**Large noise.** Using Claim A.13, for any $t \in S_i$,

$$\Pr_{\sigma,b}[E_{\mathrm{noise}}(t)] \leq 4\alpha_i.$$

By a union bound over the above three events, we have $t$ is "well isolated" with probability at least $1 - 6\alpha_i$. $\qquad\square$

**Lemma B.2.** *Given parameters $C \geq 1000$, $\gamma \leq 1/1000$. For any $k \geq 1, \epsilon \in (0,1)$, $R \geq 1$. For each $i \in [R]$, we define*

$$k_i := k\gamma^{i-1},$$
$$\epsilon_i := \epsilon(10\gamma)^i,$$
$$\alpha_i := 1/(200i^3),$$
$$B_i := C \cdot k_i/(\alpha_i^2 \epsilon_i).$$

*For each $i \in [R]$: If for all $j \leq [i-1]$ we have*

1. $\mathrm{supp}(\widehat{w}^{(j)}) \subseteq S_j$.

2. $|S_{j+1}| \leq k_{j+1}$.

3. $\widehat{z}^{(j+1)} = \widehat{z}^{(j)} + \widehat{w}^{(j)}$.

4. $\widehat{x}^{(j+1)} = \widehat{x} - \widehat{z}^{(j+1)}$.

5. $\|\widehat{x}_{\overline{S}_{j+1}}^{(j+1)}\|_2^2 \leq (1+\epsilon_j)\|\widehat{x}_{\overline{S}_j}^{(j)}\|_2^2 + \epsilon_j \delta^2 n \|\widehat{x}\|_1^2$.

*Then, with probability $1 - 10\alpha_i/\gamma$, we have*

$$|S_{i+1}| \leq k_{i+1}.$$

*Proof.* We consider a particular step $i$. We can condition on $|S_i| \leq k_i$.

By Lemma 6.1, we have $t$ is "well isolated" with probability at least $1 - 6\alpha_i$.

Therefore, each $t \in S_i$ lies in $T_i$ with probability at least $1 - 6\alpha_i$. We have Then by Markov's inequality (See Lemma A.1) and assumption in the statement, we have

$$|S_i \backslash T_i| \leq \gamma k_i \tag{3}$$

with probability $1 - 6\alpha_i/\gamma$. Then we know that

$$\begin{aligned} |S_{i+1}| &= |S_i \backslash T_i| \\ &\leq \gamma k_i \\ &\leq k_{i+1}. \end{aligned}$$

where the first step follows from the definition of $S_{i+1} = S_i \backslash T_i$, the second step follows from Eq. (3), the third step follows from the definition of $k_i$ and $k_{i+1}$.

$\square$

**Lemma B.3.** *Given parameters $C \geq 1000$, $\gamma \leq 1/1000$. For any $k \geq 1, \epsilon \in (0,1)$, $R \geq 1$. For each $i \in [R]$, we define*

$$k_i := k\gamma^{i-1},$$
$$\epsilon_i := \epsilon(10\gamma)^i,$$
$$\alpha_i := 1/(200i^3),$$
$$B_i := C \cdot k_i/(\alpha_i^2 \epsilon_i).$$

*For each $i \in [R]$: If for all $j \leq [i-1]$ we have*

1. $\text{supp}(\widehat{w}^{(j)}) \subseteq S_j$.

2. $|S_{j+1}| \leq k_{j+1}$.

3. $\widehat{z}^{(j+1)} = \widehat{z}^{(j)} + \widehat{w}^{(j)}$.

4. $\widehat{x}^{(j+1)} = \widehat{x} - \widehat{z}^{(j+1)}$.

5. $\|\widehat{x}_{\overline{S}_{j+1}}^{(j+1)}\|_2^2 \leq (1+\epsilon_j)\|\widehat{x}_{\overline{S}_j}^{(j)}\|_2^2 + \epsilon_j \delta^2 n \|\widehat{x}\|_1^2$.

*Then, with probability $1 - 10\alpha_i/\gamma$, we have*

$$\Pr\left[\left\|\widehat{x}_{T_i}^{(i)} - \widehat{w}^{(i)}\right\|_2^2 \leq \frac{\epsilon_i}{20}(\|\widehat{x}_{\overline{S}_i}^{(i)}\|_2^2 + \delta^2 n \|\widehat{x}\|_1^2)\right] \geq 1 - \alpha_i.$$

*Proof.* We define $\rho^{(i)}$ and $\mu^{(i)}$ as follows

$$\rho^{(i)} = \left\|\widehat{x}_{\overline{S}_i}^{(i)}\right\|_2^2 + \delta^2 n \|\widehat{x}\|_1^2,$$
$$\mu^{(i)} = \frac{\epsilon_i}{k_i}\left(\left\|\widehat{x}_{\overline{S}_i}^{(i)}\right\|_2^2 + \delta^2 n \|\widehat{x}\|_1^2\right). \tag{4}$$

For a fixed $t \in S_i$, let $j = h_{\sigma,b}(t)$. By Lemma A.15, we have

$$\widehat{u}_j - \widehat{x}_t^{(i)}\omega^{a\sigma t} = \sum_{t' \in T_i} \widehat{G}'_{-o_\sigma(t')}\widehat{x}_{t'}^{(i)}\omega^{a\sigma t'} \pm \delta\|\widehat{x}\|_1 \tag{5}$$

For each $t \in S_i$, we define set $Q_{i,t} = h_{\sigma,b}^{-1}(j)\backslash\{t\}$. Let $T_i$ be the set of coordinates $t \in S_i$ such that $Q_{i,t} \cap S_i = \emptyset$. Then it is easy to observe that

$$\sum_{t \in T_i}\left|\sum_{t' \in Q_{i,t}} \widehat{G}'_{-o_\sigma(t')}\widehat{x}_{t'}^{(i)}\omega^{a\sigma t'}\right|^2$$

$$= \sum_{t \in T_i}\left|\sum_{t' \in Q_{i,t}\backslash S_i} \widehat{G}'_{-o_\sigma(t')}\widehat{x}_{t'}^{(i)}\omega^{a\sigma t'}\right|^2$$

$$\leq \sum_{t \in S_i}\left|\sum_{t' \in Q_{i,t}\backslash S_i} \widehat{G}'_{-o_\sigma(t')}\widehat{x}_{t'}^{(i)}\omega^{a\sigma t'}\right|^2$$

where the first step comes from $Q_{i,t} \cap S_i = \emptyset$, and the second step follows that $T_i \subseteq S_i$.

We can calculate the expectation of $\|\widehat{x}_{T_i}^{(i)} - \widehat{w}^{(i)}\|_2^2$.

We first demonstrate that

$$\mathbb{E}_{\sigma,a,b}\left[\left\|\widehat{x}_{T_i}^{(i)} - \widehat{w}^{(i)}\right\|_2^2\right] = \mathbb{E}_{\sigma,a,b}\left[\sum_{t \in T_i}|\widehat{x}_t^{(i)} - \widehat{u}_{h_{\sigma,b}(t)}\omega^{-a\sigma t}|^2\right].$$

then get the upper bound of

$$\mathop{\mathbb{E}}_{\sigma,a,b}\left[\sum_{t\in T_i}|\widehat{x}_t^{(i)} - \widehat{u}_{h_{\sigma,b}(t)}\omega^{-a\sigma t}|^2\right]$$

.

We have

$$\mathop{\mathbb{E}}_{\sigma,a,b}\left[\left\|\widehat{x}_{T_i}^{(i)} - \widehat{w}^{(i)}\right\|_2^2\right] = \mathop{\mathbb{E}}_{\sigma,a,b}\left[\sum_{t\in T_i}|\widehat{x}_t^{(i)} - \widehat{w}_t^{(i)}|^2\right]$$

$$= \mathop{\mathbb{E}}_{\sigma,a,b}\left[\sum_{t\in T_i}|\widehat{x}_t^{(i)} - \widehat{u}_{h_{\sigma,b}(t)}\omega^{-a\sigma t}|^2\right]$$

$$= \mathop{\mathbb{E}}_{\sigma,a,b}\left[\sum_{t\in T_i}|\widehat{x}_t^{(i)}\omega^{a\sigma t} - \widehat{u}_{h_{\sigma,b}(t)}|^2\right]$$

where the first step follows that summation over $T_i$, the second step comes from the definition of $\widehat{w}_t^{(i)}$ (in Line 19 in Algorithm 1), the third step follows that

$$|\widehat{x}_t^{(i)} - \widehat{u}_{h_{\sigma,b}(t)}\omega^{-a\sigma t}| = |\omega^{-a\sigma t}| \cdot |\widehat{x}_t^{(i)}\omega^{a\sigma t} - \widehat{u}_{h_{\sigma,b}(t)}|$$

and $|\omega^{-a\sigma t}| = 1$, the fourth step comes from Eq. (5).

And then we have

$$\mathop{\mathbb{E}}_{\sigma,a,b}\left[\left\|\widehat{x}_{T_i}^{(i)} - \widehat{w}^{(i)}\right\|_2^2\right]$$

$$= \mathop{\mathbb{E}}_{\sigma,a,b}\left[\sum_{t\in T_i}|\widehat{x}_t^{(i)}\omega^{a\sigma t} - \widehat{u}_{h_{\sigma,b}(t)}|^2\right]$$

$$\leq \sum_{t\in S_i} 2\mathop{\mathbb{E}}_{\sigma,a,b}\left[\left|\sum_{t'\in Q_{i,t}\setminus S_i}\widehat{G}'_{-o_\sigma(t')}\widehat{x}_{t'}^{(i)}\omega^{a\sigma t'}\right|^2\right] + \delta^2\|\widehat{x}\|_1^2$$

$$\leq \sum_{t\in S_i} 2\mathop{\mathbb{E}}_{\sigma,b}\left[\sum_{t'\in Q_{i,t}\setminus S_i}\left|\widehat{G}'_{-o_\sigma(t')}\widehat{x}_{t'}^{(i)}\right|^2\right] + \delta^2\|\widehat{x}\|_1^2$$

$$= \sum_{t\in S_i} 2\mathop{\mathbb{E}}_{\sigma,b}\left[\sum_{t'\in\overline{S}_i}\mathbf{1}(t'\in Q_{i,t}\setminus S_i)\cdot\left|\widehat{G}'_{-o_\sigma(t')}\widehat{x}_{t'}^{(i)}\right|^2\right] + \delta^2\|\widehat{x}\|_1^2$$

$$\leq \sum_{t\in S_i}\left(\frac{1}{B_i}\|\widehat{x}_{\overline{S}_i}^{(i)}\|_2^2 + \delta^2\|\widehat{x}\|_1^2\right)$$

$$\leq \frac{|S_i|}{B_i}\|\widehat{x}_{\overline{S}_i}^{(i)}\|_2^2 + \delta^2|S_i|\cdot\|\widehat{x}\|_1^2$$

$$\leq \frac{\epsilon_i\alpha_i^2}{C}\|\widehat{x}_{\overline{S}_i}^{(i)}\|_2^2 + \delta^2|S_i|\cdot\|\widehat{x}\|_1^2,$$

where the first step follows the equation above, the second step follows Lemma 5.3, the third step follows from expanding the squared sum, the fourth step follows that if $A_1 \subseteq A_2$, we have

$$\sum_{i\in A_1} f(i) = \sum_{i\in A_2}\mathbf{1}(i\in A_1)f(i),$$

the fifth step follows for two pairwise independent random variable $t$ and $t'$, we have $h_{\sigma,b}(t) = h_{\sigma,b}(t')$ holds with probability at most $1/B_i$, the sixth step comes from the summation over $S_i$, and the last step follows from $|S_i| \leq k_i$ and $B_i = C \cdot k_i/(\alpha_i^2\epsilon_i)$.

Then, using Markov's inequality, we have,

$$\Pr\left[\left\|\widehat{x}_{T_i}^{(i)} - \widehat{w}^{(i)}\right\|_2^2 \geq \frac{\epsilon_i \alpha_i}{C}\|\widehat{x}_{\overline{S}_i}^{(i)}\|_2^2 + \delta^2 \frac{|S_i|}{\alpha_i}\|\widehat{x}\|_1^2\right] \leq \alpha_i.$$

Note that

$$\frac{\epsilon_i \alpha_i}{C}\|\widehat{x}_{\overline{S}_i}^{(i)}\|_2^2 + \delta^2 \frac{|S_i|}{\alpha_i}\|\widehat{x}\|_1^2 \leq \frac{\epsilon_i}{C}\|\widehat{x}_{\overline{S}_i}^{(i)}\|_2^2 + \delta^2 \frac{|S_i|}{\alpha_i}\|\widehat{x}\|_1^2$$

$$\leq \frac{\epsilon_i}{C}\|\widehat{x}_{\overline{S}_i}^{(i)}\|_2^2 + \frac{\epsilon_i}{C}\delta^2 B_i\|\widehat{x}\|_1^2$$

$$\leq \frac{\epsilon_i}{C}\|\widehat{x}_{\overline{S}_i}^{(i)}\|_2^2 + \frac{\epsilon_i}{C}\delta^2 n\|\widehat{x}\|_1^2$$

$$\leq \frac{\epsilon_i}{20}(\|\widehat{x}_{\overline{S}_i}^{(i)}\|_2^2 + \delta^2 n\|\widehat{x}\|_1^2),$$

where the first step follows by $\alpha_i \leq 1$, the second step follows by $|S_i| \leq k_i = \epsilon_i B_i \alpha_i^2 / C$, the third step follows by $B_i \leq n$, the last step follows by $C \geq 1000$.

Thus, we have

$$\Pr\left[\left\|\widehat{x}_{T_i}^{(i)} - \widehat{w}^{(i)}\right\|_2^2 \leq \frac{\epsilon_i}{20}(\|\widehat{x}_{\overline{S}_i}^{(i)}\|_2^2 + \delta^2 n\|\widehat{x}\|_1^2)\right] \geq 1 - \alpha_i.$$

$\square$

**Lemma B.4.** *Given parameters $C \geq 1000$, $\gamma \leq 1/1000$. For any $k \geq 1, \epsilon \in (0,1)$, $R \geq 1$. For each $i \in [R]$, we define*

$$k_i := k\gamma^{i-1},$$
$$\epsilon_i := \epsilon(10\gamma)^i,$$
$$\alpha_i := 1/(200i^3),$$
$$B_i := C \cdot k_i/(\alpha_i^2 \epsilon_i).$$

*For each $i \in [R]$: If for all $j \leq [i-1]$ we have*

1. $\operatorname{supp}(\widehat{w}^{(j)}) \subseteq S_j$.

2. $|S_{j+1}| \leq k_{j+1}$.

3. $\widehat{z}^{(j+1)} = \widehat{z}^{(j)} + \widehat{w}^{(j)}$.

4. $\widehat{x}^{(j+1)} = \widehat{x} - \widehat{z}^{(j+1)}$.

5. $\|\widehat{x}_{\overline{S}_{j+1}}^{(j+1)}\|_2^2 \leq (1+\epsilon_j)\|\widehat{x}_{\overline{S}_j}^{(j)}\|_2^2 + \epsilon_j \delta^2 n\|\widehat{x}\|_1^2$.

*Then, with probability $1 - 10\alpha_i/\gamma$, we have*

1. $\operatorname{supp}(\widehat{w}^{(i)}) \subseteq S_i$.

2. $|S_{i+1}| \leq k_{i+1}$.

3. $\widehat{z}^{(i+1)} = \widehat{z}^{(i)} + \widehat{w}^{(i)}$.

4. $\widehat{x}^{(i+1)} = \widehat{x} - \widehat{z}^{(i+1)}$.

5. $\|\widehat{x}_{\overline{S}_{i+1}}^{(i+1)}\|_2^2 \leq (1+\epsilon_i)\|\widehat{x}_{\overline{S}_i}^{(i)}\|_2^2 + \epsilon_i \delta^2 n\|\widehat{x}\|_1^2$.

*Proof.* We will prove the five results one by one.

**Part 1.**

Follows from Line 19 in the Algorithm 1, we have that

$$\text{supp}(\widehat{w}^{(i)}) \subseteq S_i.$$

**Part 2.**

By Lemma 6.2, we have that

$$|S_{i+1}| \leq k_i.$$

**Part 3.**

Follows from Line 7 in the Algorithm 1, we have that

$$\widehat{z}^{(i+1)} = \widehat{z}^{(i)} + \widehat{w}^{(i)}.$$

**Part 4.**

Follows from Line 28 in the Algorithm 1, we have that

$$\widehat{x}^{(i+1)} = \widehat{x} - \widehat{z}^{(i+1)}.$$

**Part 5.**

By Lemma B.3, we have that

$$\Pr\left[\left\|\widehat{x}_{T_i}^{(i)} - \widehat{w}^{(i)}\right\|_2^2 \leq \frac{\epsilon_i}{20}(\|\widehat{x}_{\overline{S}_i}^{(i)}\|_2^2 + \delta^2 n\|\widehat{x}\|_1^2)\right] \geq 1 - \alpha_i. \tag{6}$$

Recall that

$$\widehat{w}^{(i)} = \widehat{z}^{(i+1)} - \widehat{z}^{(i)} = \widehat{x}^{(i)} - \widehat{x}^{(i+1)}.$$

It is obvious that

$$\text{supp}(\widehat{w}^{(i)}) \subseteq T_i.$$

Conditioning on all coordinates in $T_i$ are well isolated and Eq. (6) holds, we have

$$\begin{aligned}
\|\widehat{x}_{\overline{S}_{i+1}}^{(i+1)}\|_2^2 &= \|(\widehat{x}^{(i)} - \widehat{w}^{(i)})_{\overline{S}_{i+1}}\|_2^2 \\
&= \|\widehat{x}_{\overline{S}_{i+1}}^{(i)} - \widehat{w}_{\overline{S}_{i+1}}^{(i)}\|_2^2 \\
&= \|\widehat{x}_{\overline{S}_{i+1}}^{(i)} - \widehat{w}^{(i)}\|_2^2 \\
&= \|\widehat{x}_{\overline{S}_i \cup T_i}^{(i)} - \widehat{w}^{(i)}\|_2^2 \\
&= \|\widehat{x}_{\overline{S}_i}^{(i)}\|_2^2 + \|\widehat{x}_{T_i}^{(i)} - \widehat{w}^{(i)}\|_2^2 \\
&\leq \|\widehat{x}_{\overline{S}_i}^{(i)}\|_2^2 + \epsilon_i(\|\widehat{x}_{\overline{S}_i}^{(i)}\|_2^2 + \delta^2 n\|\widehat{x}\|_1^2) \\
&= (1 + \epsilon_i)\|\widehat{x}_{\overline{S}_i}^{(i)}\|_2^2 + \epsilon_i \delta^2 n\|\widehat{x}\|_1^2.
\end{aligned}$$

where the first step comes from $\widehat{x}^{(i+1)} = \widehat{x}^{(i)} - \widehat{w}^{(i)}$, the second step is due to rearranging the terms, the third step is due to $\widehat{w}^{(i)} = \widehat{w}_{\overline{S}_{i+1}}^{(i)}$, and the fourth step comes from $S_i = T_i \cup S_{i+1}$, the fifth step is due to rearranging the terms, the sixth step the comes from a Eq. (6), and the final step comes from merging the $\|\widehat{x}_{\overline{S}_i}^{(i)}\|_2^2$ terms. $\qquad\square$

### B.2 INDUCTION TO ALL THE ITERATIONS

For completeness, we give the induced result among the all the iterations ($i \in [R]$). By the following lemma at hand, we can finally attain the theorem in Section B.3.

**Lemma B.5.** *Given parameters $C \geq 1000$, $\gamma \leq 1/1000$. For any $k \geq 1, \epsilon \in (0,1)$, $R \geq 1$. For each $i \in [R]$, we define*

$$k_i := k\gamma^{i-1},$$
$$\epsilon_i := \epsilon(10\gamma)^i,$$
$$\alpha_i := 1/(200i^3),$$
$$B_i := C \cdot k_i/(\alpha_i^2 \epsilon_i).$$

*For each $i \in [R]$, we have with probability $1 - 10\alpha_i/\gamma$, we have*

$$|S_{i+1}| \leq k_i$$

*and*

$$\|\widehat{x}_{\overline{S}_{i+1}}^{(i+1)}\|_2^2 \leq (1+\epsilon_i)\|\widehat{x}_{\overline{S}_i}^{(i)}\|_2^2 + \epsilon_i\delta^2 n\|\widehat{x}\|_1^2$$

*Proof.* Our proof can be divided into two parts. At first, we consider the correctness of the inequalities above with $i = 1$. And then based on the result we attain above (See Lemma B.4 ) and inducing over $i \in [n]$, the proof will be complete.

By Lemma 6.1, we have with probability $1 - 6\alpha_1$, $t$ is well isolated (See Definition 5.10).

**Part 1.**

We have $|S_1| = |S| \leq k = k_i$. (See Definition 3.2). And then by Lemma B.3, we have that for $i \in [R]$, $|S_{i+1}| \leq k_i$.

**Part 2.** Given that all coordinates $t \in [n]$ in $T_1$ are well isolated, with probability at least $1 - 10\alpha_i/\gamma$, we have

$$\|\widehat{x}_{\overline{S}_2}^{(2)}\|_2^2 = \|(\widehat{x}^{(1)} - \widehat{w}^{(1)})_{\overline{S}_2}\|_2^2$$
$$= \|\widehat{x}_{\overline{S}_2}^{(1)} - \widehat{w}_{\overline{S}_2}^{(1)}\|_2^2$$
$$= \|\widehat{x}_{\overline{S}_2}^{(1)} - \widehat{w}^{(1)}\|_2^2$$
$$= \|\widehat{x}_{\overline{S}_1 \cup T_1}^{(1)} - \widehat{w}^{(1)}\|_2^2$$
$$= \|\widehat{x}_{\overline{S}_1}^{(1)}\|_2^2 + \|\widehat{x}_{T_1}^{(1)} - \widehat{w}^{(1)}\|_2^2$$
$$\leq \|\widehat{x}_{\overline{S}_1}^{(1)}\|_2^2 + \epsilon_1(\|\widehat{x}_{\overline{S}_1}^{(1)}\|_2^2 + \delta^2 n\|\widehat{x}\|_1^2)$$
$$= (1+\epsilon_1)\|\widehat{x}_{\overline{S}_1}^{(1)}\|_2^2 + \epsilon_1\delta^2 n\|\widehat{x}\|_1^2.$$

where the first step comes from $\widehat{x}^{(2)} = \widehat{x}^{(1)} - \widehat{w}^{(1)}$, the second step is due to rearranging the terms, the third step is due to $\widehat{w}^{(1)} = \widehat{w}_{\overline{S}_2}^{(1)}$, and the forth step comes from $S_1 = T_1 \cup S_2$, the fifth step is due to rearranging the terms, the sixth step the comes from expanding the terms, and the final step comes from merging the $\|\widehat{x}_{\overline{S}_1}^{(1)}\|_2^2$ terms.

By Lemma B.4, for all $i \in [R]$, we can have

$$\|\widehat{x}_{\overline{S}_{i+1}}^{(i+1)}\|_2^2 \leq (1+\epsilon_i)\|\widehat{x}_{\overline{S}_i}^{(i)}\|_2^2 + \epsilon_i\delta^2 n\|\widehat{x}\|_1^2$$

$\square$

## B.3 MAIN RESULT

In this subsection, we give the main result as the following theorem.

**Theorem B.6** (Main result). *If all of the requirements are met*

- **Requirement 1.** *Let $x \in \mathbb{C}$.*

- **Requirement 2.** *We denote $\widehat{x}$ as the Fourier transformation result.*

- **Requirement 3.** *Let $\epsilon \in (0,1)$, $\delta \in (0,1)$.*

- **Requirement 4.** *We define $S \subseteq [n]$, $|S| = k$ where $k \geq 1$.*

*An algorithm (Algorithm 1) exists such that*

- **Part 1.** *It takes $O(\epsilon^{-1} k \log(n/\delta))$ samples from $x$.*

- **Part 2.** *It runs in $O(\epsilon^{-1} k \log(n/\delta))$.*

- **Part 3.** *It holds with probability at least $9/10$.*

- **Part 4.** *It outputs a vector $x' \in \mathbb{C}^n$ such that*

$$\|(x' - \widehat{x})_S\|_2^2 \leq \epsilon \|\widehat{x}_{\overline{S}}\|_2^2 + \delta \|\widehat{x}\|_1^2,$$

*Proof.* By Lemma 6.4, we can conclude that with $R = \log k$ iterations, we will obtain the result we want. Then we will give the analysis about the time complexity and sample complexity.

**Proof of Part 1.** From analysis above, the sample needed in each iteration is $O((B_i/\alpha_i) \log(n/\delta))$ then we have the following complexity.

The sample complexity of ESTIMATION is

$$\sum_{i=1}^{R} (B_i/\alpha_i) \log(n/\delta) = O(\epsilon^{-1} k \log(n/\delta)).$$

**Proof of Part 2.** The time in each iteration mainly from two parts. The EstimateValues and HashToBins functions. For the running time of EstimateValues, its running time is mainly from loop. The number of the iterations of the loop can be bounded by

$$O(B_i/\alpha_i \log(n/\delta))$$

.

By Lemma A.15, we can obtain the time complexity of HashToBins with the bound of

$$O(B_i/\alpha_i \log(n/\delta)).$$

This function is used only once at each iteration.

Let $R = \log k$. We can have the following equation. The Time complexity of ESTIMATION is

$$\sum_{i=1}^{R} (B_i/\alpha_i) \log(n/\delta) = O(\epsilon^{-1} k \log(n/\delta)).$$

**Proof of Part 3.** We union bound the query error probability over the iterations $R = \log k$ in Lemma B.4.

Using Lemma B.4, we can obtain the failure probability in each iteration as $alpha_i/\gamma$.

Thus, the overall failure probability can be expressed as follows:

$$\sum_{i=1}^{R} 10\alpha_i/\gamma < 1/10.$$

**Proof of Part 4.** To bound the query error, we will bound the $\|\widehat{x}_{\overline{S}_i}^{(i)}\|_2^2$ first. By Lemma B.4, it follows that

$$\|\widehat{x}_{\overline{S}_i}^{(i)}\|_2^2 \leq (1 + \epsilon_i)\|\widehat{x}_{\overline{S}_i}^{(i)}\|_2^2 + \epsilon_i \delta^2 n \|\widehat{x}\|_1^2$$

$$
\begin{aligned}
&\leq (1+\epsilon_i)(1+\epsilon_{i-1})\|\widehat{x}_{\overline{S}_{i-1}}^{(i-1)}\|_2^2 \\
&\quad + ((1+\epsilon_i)\epsilon_{i-1} + \epsilon_i)\delta^2 n\|\widehat{x}\|_1^2 \\
&\leq \prod_{j=1}^{i}(1+\epsilon_j)\|\widehat{x}_{\overline{S}_j}\|_2^2 + \sum_{j=1}^{i}\epsilon_j\delta^2 n\|\widehat{x}\|_1^2 \prod_{l=j+1}^{i}(1+\epsilon_l) \\
&\leq 8(\|\widehat{x}_{\overline{S}_i}\|_2^2 + \delta^2 n\|\widehat{x}\|_1^2),
\end{aligned}
\tag{7}
$$

where the first step comes from the assumption in Lemma B.4, the second step comes from the assumption in Lemma B.4, the third step refers to recursively applying the second step, the last step follows from simple algebra.

Now the query error can be bounded as follows

$$
\begin{aligned}
\|\widehat{x}_S - \widehat{z}^{(R+1)}\|_2^2 &= \sum_{i=1}^{R}\|\widehat{x}_{T_i}^{(i)} - \widehat{w}^{(i)}\|_2^2 \\
&\leq \sum_{i=1}^{R} k_i \mu^{(i)}/20 \\
&\leq \sum_{i=1}^{R} \epsilon_i(\|\widehat{x}_{\overline{S}_i}^{(i)}\|_2^2 + \delta^2 n\|\widehat{x}\|_1^2)/20 \\
&\leq \sum_{i=1}^{R} \epsilon_i \cdot 10(\|\widehat{x}_{\overline{S}}\|_2^2 + \delta^2 n\|\widehat{x}\|_1^2)/20 \\
&\leq \epsilon(\|\widehat{x}_{\overline{S}}\|_2^2 + \delta^2 n\|\widehat{x}\|_1^2).
\end{aligned}
$$

where the first step follows that $T_i$ is well isolated (See Definition 5.10) and $\widehat{w}^{(i)} = \widehat{z}^{(i+1)} - \widehat{z}^{(i)}$, the second step is by Eq. (6), the third step comes from definition of $\mu^{(i)}$ in Eq. (4), the fourth step follows from Eq.(7), and the final step follows from the geometric sum, $\epsilon_i = \epsilon(10\gamma)^i$ and $\gamma \leq 1/1000$. $\quad\square$

