# OpenReview forum: "An $O(k\log n)$ Time Fourier Set Query Algorithm"
_ICLR.cc/2025/Conference — Submitted to ICLR 2025_

### Official Review · Reviewer_fimC · 2024-10-27

**Soundness:** 3
**Presentation:** 3
**Contribution:** 3
**Rating:** 8
**Confidence:** 3

**Summary:**

This paper studies the design of sublinear time and query algorithms for the Fourier set query problem. In this problem, one is given query access to the entries of $x\in \mathbb{C}^n$, a set $S\subseteq [n]$ of size $k$ and parameters $\epsilon,\delta >0$. The goal is to output a vector $x'$ such that $\|(x'-\hat{x})_S\|_2^2 \leq \epsilon \|(\hat{x})_T\|_2^2  +\delta \|\hat{x}\|_1^2$ where $\hat{x}$ is the Fourier transform of $x$ and $T=[n]\setminus S$. The previous best algorithms for this problem had query complexity $\epsilon^{-1}k$ and runtime $\epsilon^{-1}k \log^{2.1}n \log (R^*)$ ($R^*$ is the infinity norm of $x$) or query complexity $\epsilon^{-1}k\log^2 n$ and runtime $\epsilon^{-1}k \log^{2}n $. The contribution of this paper is to design an algorithm with $\epsilon^{-1}k \log n$ sample complexity and runtime.

**Strengths:**

The main strengths of the paper are to improve both the sample and time complexity of the original algorithm of Hassanieh et al. using core techniques, such as using HashToBins to isolate frequencies, recover them and iterate, that originate from Hassanieh et al. directly. This thus shows that existing techniques can indeed achieve better results. The algorithm also improves upon the algorithm of Kapralov et. al. in terms of the time complexity by improving log factors and removing dependence on the infinity norm of the vector.

**Weaknesses:**

One weakness of the algorithm is that it does not explain how the application of the classical techniques in Sublinear Fourier analysis such as HashToBins and iterative frequency recovery are applied differently in this paper compared to existing works to ensure the improvement in the result. Moreover a minor technical limitation of the paper is to obtain worse sample complexity compared to the algorithm of Kapralov, however this is not a big limitation to discredit the novelty of the algorithm in applying existing techniques to refine the algorithmic guarantees.

**Questions:**

Can the authors please explain briefly what is the difference in the approach of this algorithm, with regards to using HashToBins and iterative frequency recovery after isolating frequencies in each round of hashing and recovering, compared to Hassanieh et al and Kapralov's result ?

---

> ### Author Response · Authors · 2024-11-23
> **Thank you very much for your insightful comments!**
>
> We express our deepest gratitude to the reviewer for these insightful comments. Below, we respond to the weaknesses to address your concerns:
>
> Our algorithm achieves $O(\epsilon^{-1} k \log(n/\delta))$ sample complexity compared to Kapralov's $O(k/\epsilon)$
> This trade-off enables simpler implementation with fewer parameters and better runtime dependency by eliminating $\log R^*$ factor.
>
> For many practical applications, the additional $\log(n/\delta)$ factor in sample complexity is acceptable given the runtime improvements.
>
> Thank you again for your insightful comments.

---

### Official Review · Reviewer_V6Eq · 2024-10-29

**Soundness:** 2
**Presentation:** 2
**Contribution:** 3
**Rating:** 3
**Confidence:** 3

**Summary:**

Fourier set query is similar in nature to sparse FFTs, except in this manuscript the sparsity set is selected a priori. The algorithm has both sample and time complexity of O(eps^{-1}klog(n/delta)), and guarantees an eps/delta-accurate set of Fourier modes in the query set S with high-probability.

**Strengths:**

The manuscript looks original and it appears to have improved on both the sample and time complexity of the set query Fourier problem.

**Weaknesses:**

- The main techniques look similar to those found in Hassanieh et. al. I believe it is mostly technique II that is different from Hassanieh et. al. This comes across as a very carefully performed iteration on the set S, which seems tricky and technical.  From the manuscript, I wasn’t able to follow the reasoning to convince myself that the manuscript is technically correct.

- The manuscript is poorly written in my opinion. Having informal versions of many of the lemmas is not very helpful. Instead, it would have been better if the manuscript used that space to explain the many technical contribution of the manuscript.

- I really wished the manuscript had a few sentences about the motivational applications. I am used to sparse Fourier problem, where the query set S is not known in advance, and the algorithm is designed to get both a set S (with |S|=k) and the Fourier modes in S. In that setting, the Hassanieh et. al paper from 2012 takes O(k*log(n)*log(n/k)) in samples and running time. It is also a deterministic algorithm. Query set Fourier, on the other hand, is limited to retrieving specified frequencies without leveraging the full sparsity of the signal.

**Questions:**

- Can you give me a few motivating applications where the set query Fourier problem appears, not the sparse Fourier problem?

- How does your algorithm compare to Hassanieh et. al from 2012?

- Would it be possible to implement your set query Fourier algorithm?  How does it behave under floating-point rounding errors?

---

> ### Author Response · Authors · 2024-11-23
> **Thank you very much for your insightful comments!**
>
> We express our deepest gratitude to the reviewer for these insightful comments. Below, we respond to the weaknesses to address your concerns:
>
> While we do build upon Hassanieh et al.'s foundation, our contribution introduces several crucial technical differences, we give a novel iterative compression scheme for the query set $S$ that achieves $O(k \log n)$ complexity by carefully managing the interaction between successive iterations. We provide a refined analysis of the error propagation that eliminates the extra $\log n$ factor present in prior work and a new proof technique for bounding the interaction between isolated frequencies across iterations.
>
> Thank you again for your insightful comments.

---

> > ### Comment · Reviewer_V6Eq · 2024-11-25
> > **Acknowledgment of reply**
> >
> > Thank you for your response. I asked three questions in my referee report, which were not explicitly answered in the reply.  My score and opinion of the manuscript remains unchanged.

---

### Official Review · Reviewer_SsZL · 2024-11-03

**Soundness:** 2
**Presentation:** 1
**Contribution:** 1
**Rating:** 1
**Confidence:** 5

**Summary:**

This paper propose an algorithm to compute the approximate Fourier transform on a query set of size $k$. Through the techniques of HASHTOBINS, they obtain the error analysis between  the approximate Fourier transform and Fourier transform,

**Strengths:**

NA

**Weaknesses:**

1. This paper is poorly written & presented. A lot of the content can be found in the undergraduate textbook. A substantial part of the results are informal version, say Lemma 6.1 - 6.3. Also, there is hardly any interpretation of the main results. The presentation style does not seem to be serious.
2. The technical contribution is unclear. Most of the analysis are quite standard.
3. There is no numerical experiments to verify its application in real-world dataset.

**Questions:**

See above

---

> ### Author Response · Authors · 2024-11-23
> **Thank you very much for your insightful comments!**
>
> We express our deepest gratitude to the reviewer for these insightful comments. Below, we respond to the weaknesses to address your concerns:
>
> We have several key technical innovations in our work. First, we present a novel approach to handling the Fourier set query problem that reduces time complexity from $O(k \log^2 n)$ to $O(k \log n)$.
> We introduce a more efficient hashing scheme that eliminates the dependence on $\log R^*$ present in previous work.
> We develop new proof techniques for bounding the error probability in the iterative loop analysis. These improvements, while utilizing some standard analytical tools, combine to create a significant advancement in the field.
>
> Thank you again for your insightful comments.

---

### Official Review · Reviewer_sP6c · 2024-11-03

**Soundness:** 2
**Presentation:** 1
**Contribution:** 2
**Rating:** 3
**Confidence:** 4

**Summary:**

This paper studies the Fourier set query problem: given a set of indices $S \subset [n]$ and a vector $x \in \mathbb{C}^n$, the goal is to compute the Fourier transform $\widehat{x}$ of $x$ at the set of indices $S$ with precision specified by parameters $\epsilon$ and $\delta$. More formally, the algorithm is required to output a vector $x'$ such that $$|| (x' - \widehat{x})_S ||^2_2 \leq \epsilon ||\widehat{x}_S||^2_2 + \delta || \widehat{x} ||^2_1. $$

Since the complete Fourier transform can be computed in time $O(n \log n)$, it is desired that the running time of the algorithm would be less than this. At the same time, for some applications the sample complexity, i.e. the number of entries in $x$ that are read by the algorithm, is also crucial.

The main contribution of this paper is an algorithm that solves the Fourier set query problem with time and sample complexity $O(\frac{k}{\epsilon} \log n)$, achieving an improvement over previous work.

**Strengths:**

The authors prove all of the made claims, including the main result. Detailed proofs for all of the original theorems and lemmas can be found in the appendix.

**Weaknesses:**

The greatest weakness of the paper is the lack of clarity, which greatly impedes assesment of its other merits. Some of the concerns are:

1. This paper uses a lot of prior work, and it is not made clear what the improvements in techniques are, or sometimes what even are the techniques. For example, in section 4 the paragraphs discussing techniques 3 and 4 consist mainly of an outline of the proof. Concrete techniques are not highlighted, and it is hard to understand which steps are standard  in the literature, and which ones are the contribution of the paper.
As far as I am aware, the only thing stated as original contribution is the algorithm itself and the technique 2.

I think that the paper should be rewritten in a way such that the key critical insights that led to the improvement of the algorithm become identifiable. My suggestions are
 - Extract the techniques 2 and 3 from the proof structure to show what do they allow you to do, preferably by formatting them as a lemma or a theorem. Otherwise, do not call those paragraphs "techniques", just a "proof overview".
 - For techniques 2 and 3, explicitly mention if the proof steps are an entirely novel contribution, and if not list works where they were borrowed from and discuss how your use of them is different from prior work.
2. Explanation of how the algorithm or proofs work is lacking intuition. The majority of the paper, starting from section 3, is made up of dry and formal derivation of the main result.
I would suggest:
 - Adding informal but detailed explanations of what each section does at its start, starting with section 3.
 - Add brief "intuition paragraphs" after each major theorem or lemma to explain its significance in plain language

As a consequence, in my opinion the significance of this paper in its current state is rather low. While the claimed result appears to be interesting, I doubt that the paper itself right now would of use for anyone, as it is very difficult to understand.

**Questions:**

I have previously reviewed this paper for a different conference, and as far as I can tell the only difference between the two versions is in Section 4, where some explanatory text have been added to techniques. In my opinion, it is a step in the right direction, but it does not resolve the concerns that I have raised back then about the readability of the paper.
In the rebuttal for the previous conference, the authors also mentioned in their reply specific steps that they would take to improve the readability, which have been partially implemented in Section 4. I think that doing the rest of them will be greatly beneficial for the state of the paper.

---

> ### Author Response · Authors · 2024-11-23
> **Thank you very much for your insightful comments!**
>
> We express our deepest gratitude to the reviewer for these insightful comments. We thank you very much for your time in reviewing our work multiple times. We will definitely keep building on the directions promised before. We have attached these directions below:
>
> $\bullet$ To present a clear comparison between our work and previous works, we will expand our technique overview section by adding more intuition and explanations of the novel techniques we have developed. Our main technical contributions include a new iterative estimation procedure that allows for faster runtime, improved analysis of the “well-isolated” property for coordinates, and refined bounds on estimation errors across iterations. We will revise the paper to explicitly highlight these contributions throughout.
>
> $\bullet$ Regarding our intuition, we will add an informal algorithm overview section that explains the key ideas before diving into technical details. We will provide intuitive explanations for important lemmas and proof steps, and include more illustrative examples to build understanding, rather than listing too many technical formulas, especially in the main text. We will revise Section 4 to give clearer descriptions of key techniques and functions before presenting formulas. For example, for Technique I, we will first explain at a high level what HashToBins and EstimateValues do conceptually, before presenting the technical details. Then, in Technique II and Technique III, where we present our novel techniques for the “well-isolated” property and the iterative estimation procedure using induction, respectively, we will expand on them by explaining how these innovations contribute to our improved results. Moreover, we are committed to presenting a more comprehensive notation section (see details from our rebuttal to Reviewer snns and Reviewer Ei5x). We believe that implementing these changes will significantly enhance the paper's clarity, readability, and impact.
>
> $\bullet$ Regarding your question, we do not think it is a typo: our goal is to improve the running time of Algorithm 2 in [1], where they use Theorem 1.1 to present its main properties. In Theorem 1.1 and its proof, they indeed have $O(k/\epsilon)$ as sample complexity.
>
> Thank you again for your insightful comments.
>
> [1] Kapralov, Michael. "Sample efficient estimation and recovery in sparse FFT via isolation on average." FOCS’17.

---

> > ### Comment · Reviewer_sP6c · 2024-11-25
> >
> > Thank you for your reply.

---

### Meta-Review · Area_Chair_BB6D · 2024-12-11

**Metareview:**

The reviewers seem to view this work as having potential, but there are multiple major concerns with the writing quality, doubts about which parts are novel vs existing (or even textbook material), and lack of intuition/motivation.  Some specific concerns/suggestions include:
- Try to improve the clarity on what is new vs existing throughout the entire paper
- Revise things like notation, captions, etc. as suggested
- Revise the problem motivation, ideally with a clear use case or application
- Try to avoid informal statements/reasoning unless there is a good justification for doing so
- Consider including experiments, though this is not strictly necessary

Based on the above, at the very least, a major revision seems to be needed.

**Additional Comments On Reviewer Discussion:**

The author response was very brief and did not seem to alleviate the main concerns.

---

### Decision · Program_Chairs · 2025-01-22

Reject